# Meta-research: justifying career disruption in funding applications, a survey of Australian researchers

Adrian Barnett[1,2]*, Katie Page[3], Carly Dyer[2], Susanna Cramb[2]

[1]School of Public Health and Social Work, Queensland University of Technology, Brisbane, Australia; [2]Australian Centre for Health Services Innovation and Centre for Healthcare Transformation, Queensland University of Technology, Brisbane, Australia; [3]Centre for Health Economics Research and Evaluation, University of Technology Sydney, Sydney, Australia

## Abstract

**Background:** When researchers' careers are disrupted by life events – such as illness or childbirth – they often need to take extended time off. This creates a gap in their research output that can reduce their chances of winning funding. In Australia, applicants can disclose their career disruptions and peer reviewers are instructed to make appropriate adjustments. However, it is not clear if and how applicants use career disruption sections or how reviewers adjust and if they do it consistently.

**Methods:** To examine career disruption, we used surveys of the Australian health and medical research community. We used both a random sample of Australian authors on *PubMed* and a non-random convenience sample.

**Results:** Respondents expressed concerns that sharing information on career disruption would harm their chances of being funded, with 13% saying they have medical or social circumstances but would not include it in their application, with concerns about appearing 'weak'. Women were more reluctant to include disruption. There was inconsistency in how disruption was adjusted for, with less time given for those with depression compared with caring responsibilities, and less time given for those who did not provide medical details of their disruption.

**Conclusions:** The current system is likely not adequately adjusting for career disruption and this may help explain the ongoing funding gap for senior women in Australia.

**Funding:** National Health and Medical Research Council Senior Research Fellowship (Barnett).

*For correspondence: a.barnett@qut.edu.au

## Editor's evaluation

This study examined the reporting of career disruption for government funding applications by the Australian research community. Through a survey, the authors found that despite a substantial number of respondents having medical or social circumstances that would be considered under career disruption, most researchers expressed concerns about sharing career disruption information with the view that it would harm their chances of being funded. The outcomes highlight the perceived stigma for reporting career disruption and impacting women to a greater degree, as well as the inadequacy of the system to foster transparency probably due to the competitive research culture.

**eLife digest** Science is an expensive endeavor. To pursue their ideas, most researchers need to win funding by submitting applications to highly competitive schemes with low success rates. Funding decisions depend on many factors, but usually take into consideration a researcher's track record: publications, collaborations with other researchers and even other awards they have received.

Researchers whose careers have been disrupted by life events, including childbearing or being ill, may have a gap in their track record that reduces their chances of winning funding. Historically, female researchers have experienced career disruptions more often, leading to a funding gap between male and female researchers. To increase fairness and reduce this gap, many funding agencies have instructed the peer reviewers – other scientists – who assess funding applications to adjust their scores to account for career disruptions. However, large funding gaps are still frequently observed between female and male researchers.

Barnett et al. wanted to know how career disruption is considered in practice by establishing what personal details are shared in applications by researchers with disruption, and how reviewers treat this information. To find out, they surveyed medical researchers in Australia and asked them for their views on career disruption as both funding applicants and reviewers of funding applications.

The answers to the survey indicated that 13% of the applicants responding had experienced career disruptions, but would not include them in funding applications. In many cases, this reluctance to disclose career disruptions was due to concerns that it would harm an applicant's chances of winning funding, a concern that was greater in the women who responded to the survey. Researchers who answered the survey would claim less time off on average if their career disruption was for severe depression compared with caring for a child or elderly relative. Additionally, the answers to the survey show that, on average, peer reviewers – the scientists who assessed the applications – would give more time off to applicants who provided details about the medical issues that caused a career disruption than to those who did not.

The results of this survey suggest that changes in the systems used to apply for funding and in how applications are assessed could make funding fairer. One suggestion would be to modify funding applications to make disruptions easier to report. Another would be to make changes to the reviewing procedures to increase privacy and reduce variability in how disruption is assessed. Changes in these directions could help researchers gain access to funding more fairly, increasing the quality and output of scientific research.

## Introduction

Obtaining funding for health and medical research has become harder in Australia as success rates have declined. Declining success rates increase the competition for funding and so create additional pressure for researchers to have an outstanding track record with large numbers of publications (*Lipton, 2020*). Gaps in track record due to career disruption can therefore be highly prejudicial when researchers are compared (*Lipton, 2020*). Researchers who experience career disruption could have a gap in their track record if they took time away from research because of a personal issue, for example, their own illness or that of a family member.

Australian funding systems rely heavily on researchers' track records (*Mow, 2009*; *Coveney et al., 2017*), meaning researchers could be unsuccessful in funding applications because of personal circumstances beyond their control. Recently, a disabled Australian researcher demonstrated that the National Health and Medical Research Council (NHMRC) peer review system was not adequately comparing their track record with their peers (*Clifford and James, 2020*). The NHMRC are the largest public funder of health and medical research in Australia.

A common reason for career disruption is caring for children, and hence career disruption is an issue often keenly felt by female researchers. There is a persistent shortfall of women awarded NHMRC funding at senior levels (*NHMRC, 2018*; *Women's Agenda, 2021*) with a recent call for an overhaul to the funding system to address this gap (*ManelWatchAu, 2021*). The NHMRC has a gender equity strategy aimed at reducing the gap (*NHMRC, 2018*). Insufficient adjustment for career disruption when assessing funding applications could be one of the causes of the funding gap for women, with

more women quitting research in mid-career when caring for children, leaving fewer women to apply at senior levels (*Resmini, 2016*; *Ysseldyk et al., 2019*).

The NHMRC allows applicants to include prolonged career disruptions which may have impacted their research performance, including child birth and carers' responsibilities, and calamities such as bushfires and the COVID-19 pandemic (*NHMRC, 2021b*). This aims to create a more equitable comparison between applicants. The NHMRC have also recently introduced a career context section, where issues that have effected research productivity but would not be considered a career disruption can be included. There is also a 'relative to opportunity' section where applicants can detail other personal or professional circumstances affecting research productivity (*NHMRC, 2021b*). The NHMRC acknowledges the potential impact of career disruptions, and instructs peer reviewers to assess research outputs relative to opportunity, but it is unclear how exactly this is achieved. It is not clear whether any adjustments are consistently applied as the assessment of career disruption is made by peer reviewers who can have widely different experiences and attitudes about career disruption (*Barnett, 2020*).

The current NHMRC funding application systems mean that researchers claiming career disruption often need to share highly personal information. For team applications, this means their colleagues can read their personal information. This information is also available to peer reviewers, who may be colleagues or potential future employers. Some researchers may therefore be reluctant to share their personal information and hence remain disadvantaged by their career disruption (*Brown and Leigh, 2018*). The extent to which this is happening is currently unknown and if this varies by gender then this could contribute to the gender funding gap.

To gather information on how Australian researchers are writing about career disruption, we designed a survey that aimed to gather information on researchers' experiences and views on dealing with career disruption when applying for funding. Our study concerns career disruption in relation to applying for funding and does not consider the similar issues experienced in job and promotion applications.

The National Health and Medical Research Council (NHMRC) are the largest government funding agency for health and medical research in Australia. In 2020–21, there were 536 grants awarded with a total budget of $497 million AUD (*NHMRC, 2021a*). Across all schemes the success percentages by the gender of the lead investigator were 13.0% for women and 12.9% for men, but were 12.7% (women) and 14.0% (men) for Investigator grants (akin to fellowships). For the lead investigators for all schemes, there were 1678 applications submitted by women and 2154 by men. There were 64 postgraduate scholarships awarded to early career researchers. Smaller external and internal schemes are available, however, promotion often requires a large grant from the NHMRC or equivalent (*Rice et al., 2020*).

## Materials and methods

We prespecified our data collection methods and analysis in a protocol (*Barnett et al., 2021a*). Ethics approval was obtained from the Queensland University of Technology human research ethics committee (LR 2021-4303-5402). Our target population was current Australian health and medical researchers, which we sampled using both a random sample and a non-random sample.

### Random sample

We used a sampling frame of emails extracted from publications on the *PubMed* database, which is a widely used search engine that contains the MEDLINE database of published papers in life sciences and biomedical topics. This is a large database and there are over 1.6 million papers with a publication year of 2020. The database is run by the US National Library of Medicine. A recent survey targeting Australian researchers used this same sampling frame (*Scott et al., 2021*).

To capture current Australian researchers we searched *PubMed* using the search query:
(2020[PDAT] OR 2021[PDAT]) AND (Australia[AFFL] OR Australian[AFFL]) AND *type*[PTYP]

Where PDAT = publication date, AFFL = affiliation, and PTYP = publication type. The publication types are listed in Appendix 1, and these were selected to focus on published papers and exclude non-research papers like obituaries. We limited the publication dates from 2020 to the latest available date to attempt to capture currently active researchers.

We downloaded all available names and emails from this search and excluded *Twitter* handles. We excluded students based on the text 'hdr', 'student', or 'postgrad' in their email; this was to focus on researchers who were more likely to have experience with applying for NHMRC funding. We only included emails that contained '.au' to limit the sampling frame to researchers with an Australian affiliation. However, this does exclude Australian researchers that use an alternative email service such *gmail*. The *PubMed* database was searched on 5 October 2021 and provided 7588 unique researchers.

## Non-random sample

A random sample is an ideal method for providing data that are representative of the target population. However, survey response rates are generally low and have declined in recent years (*Bednall et al., 2013*; *Kennedy and Hartig, 2019*), meaning getting an adequate sample size can be challenging. We therefore supplemented our random sample with a non-random convenience sample using a survey link that was openly available.

A benefit of including this non-random sample is that researchers often have strong views about funding systems (*Herbert et al., 2014*), and so allowing any Australian researcher to complete the survey will mean their opinions can be heard. It is possible that those completing the non-random survey were motivated by bad experiences concerning their own career disruption, and hence we expected the two samples to differ in terms of the prevalence of negative attitudes towards the current system.

The survey for the non-random sample included two additional questions compared with the random sample, which were: (1) for the respondent to confirm that they were a health and medical researcher working in Australia and not a current student, (2) their email, so that those who were part of the random sample could be moved over to that sample – this is worthwhile given that spam filters may have trapped our invite.

## Survey distribution

The random sample was sent in individual emails by investigator Barnett on 10 October 2021. The completion page of the survey included a link to the non-random sample to share with colleagues. The link to the non-random sample was also shared on *Twitter* and *LinkedIn* on 10 October 2021. It was also distributed in relevant departments and via a health research mailing list. Reminder emails were sent to the random sample who had not withdrawn or responded on the 17th and 21st of October (some responders wrongly received a reminder). The survey was open for 21 days.

The first page of the survey was a participant information sheet and respondents had to give their consent before they saw the questions. The survey was voluntary and there were no incentives to participate. The survey was online and used the *Qualtrics* program that allowed easy viewing on screens and smart phones.

## Sample size

We had no primary hypothesis to base a sample size calculation on, instead we aimed for a margin of error of 10% for all categorical questions, for example, yes/no, agree/neutral/disagree. The largest possible variance for a categorical question is for a proportion of 50%. So using a 95% confidence interval (CI) we would need 96 respondents to give a margin of error of 10% or less.

We assumed a response rate of 25% for our random sample, hence we inflated our target sample size by a factor of four and approached 4 × 96 = 384 researchers, who were randomly selected from the sampling frame. Researchers were selected using a random number generator in *R* created by investigator Barnett.

Our assumed response rate was based on related surveys. A survey of Australian researchers that took an average of 10 min to complete had a response rate of 21% (*Scott et al., 2021*). A short survey of Australian researchers by our team using just 1–2 questions had a response rate of 59% (*Sewell and Barnett, 2019*).

There was no sample size calculation for the non-random sample and we instead aimed to get a similar response number to the random sample.

## Survey questions

The participant information sheet and survey questions are in *Supplementary file 1*. The survey had 14 closed questions and 9 open questions over 8–9 screens. All questions could be skipped (except the consent question) and the open questions were labelled as 'optional' and were for any additional comments. There was no check of survey completeness. Respondents could move backwards to change previous answers. We did not use information about IP addresses as we reasoned that Australian researchers could be overseas.

Our questions concerned prolonged career disruption rather than 'relative to opportunity' which accounts for time spent away from research due to activities like clinical work and teaching. The sections of the survey were (in order):

- Respondents' recent application activity and knowledge of career disruption policy
- How they would approach a current NHMRC application in terms of their own career disruption
- A hypothetical scenario on career disruption
- How they approach career disruption as a peer reviewer
- Their opinion on an independent medical panel for assessing career disruption which provides a report on the time lost to the peer reviewers without the need for applicants to share personal details with colleagues or reviewers (*Barnett, 2020*)
- Basic demographics: gender, broad career field, career stage

To reduce the length of the survey, four related hypothetical scenarios were randomized so that each respondent saw only one scenario. Respondents were asked if they would write anything in the career disruption section and how much time away from research they would claim. In every scenario, we told the researcher that they had lost 6 months and only the condition changed.

We were not aware of any established survey questions on career disruption. Hence, all our questions were designed by the study team. All members of the study team have personal experience of considering what to write in career disruption sections and have considered the difficulties therein. The questions were piloted with around 20 Australian health and medical researchers, including those who have experienced career disruption.

## Statistical methods

Most results are presented for each sample (random or non-random) rather than combining them. We give the number of survey respondents in each group and the response rate for the random survey.

Results are summarized using numbers and percentages in tables and bar plots for categorical variables. Key percentages, such as the percent not reporting career disruption, were given with 95% CIs for the random sample to estimate the percentage in the target population. The random sample should be more generalizable to the target population than the non-random sample, although it is still prone to non-response bias. We did not use any survey weights as we did not have information such as age and gender for our sampling frame.

To compare our sample to the target population we extracted publicly available data from the NHMRC on the breakdown of applicants by gender and basic research area.

For the four hypothetical scenarios, we modelled the mean time away from research by assigning 0 months to those who said they would not claim any disruption and combined this with the time assigned by those who would claim disruption. We used a Bayesian model to estimate the mean for each scenario, the 95% credible interval for the mean, and the probability that the scenario had the lowest mean. As each respondent only saw one of the four scenarios we were concerned about the between group comparisons being too uncertain. Hence, this model combined the results from the random and non-random sample and included the sample as an independent variable.

We tabulated the number of item-missing responses and highlighted particular questions that were frequently skipped. We used partially completed surveys, except those that only completed the demographics section.

In a sensitivity analysis, we imputed missing data for the 'slider' questions where researchers had to select the time on a sliding range from 0 to 12 months. The starting position for these sliders was 6 months which matched the career disruption in the scenarios that were 6 months. However, respondents had to touch the slider for the 6 months to be registered, hence in a sensitivity analysis we imputed 6 months as some respondents may have agreed with the 6-month figure without clicking the slider.

In a planned subgroup analysis, we examined if female respondents were more or less likely to agree to selected questions. We used ordinal regression for responses on an ordinal scale, such as disagree, neutral and agree, and used logistic regression for responses on an nominal scale and selected a particular response, for example, support for the medical panel.

Results are reported using the CHERRIES checklist for online surveys (*Eysenbach, 2004*). We used the *R* software version 4.1.1 for data management and analysis (*R Development Core Team, 2020*).

We did not use a formal qualitative analysis of the optional comments from participants. Instead the study team read all 414 comments to identify those that explained quantitative results. We looked for comments that reflected any apparent consensus alongside other interesting or divergent views. For each comment, we include the respondent's gender and years of experience.

### Data and code availability

The analysis code and data to replicate all parts of the analyses and generate the figures and tables are available from GitHub: https://github.com/agbarnett/career_disruption, (copy archived at swh:1:rev:555bffb51ede3af1511a4707ce35aec87785caa2; *Barnett, 2021b*).

## Results

The final response percentage was 32% (124/384), which was above our target percentage of 25%. Twenty-nine emails had an 'out of office' response and seven of these mentioned they were away on maternity, parental leave or due to illness. Hence, our survey missed some of the very researchers

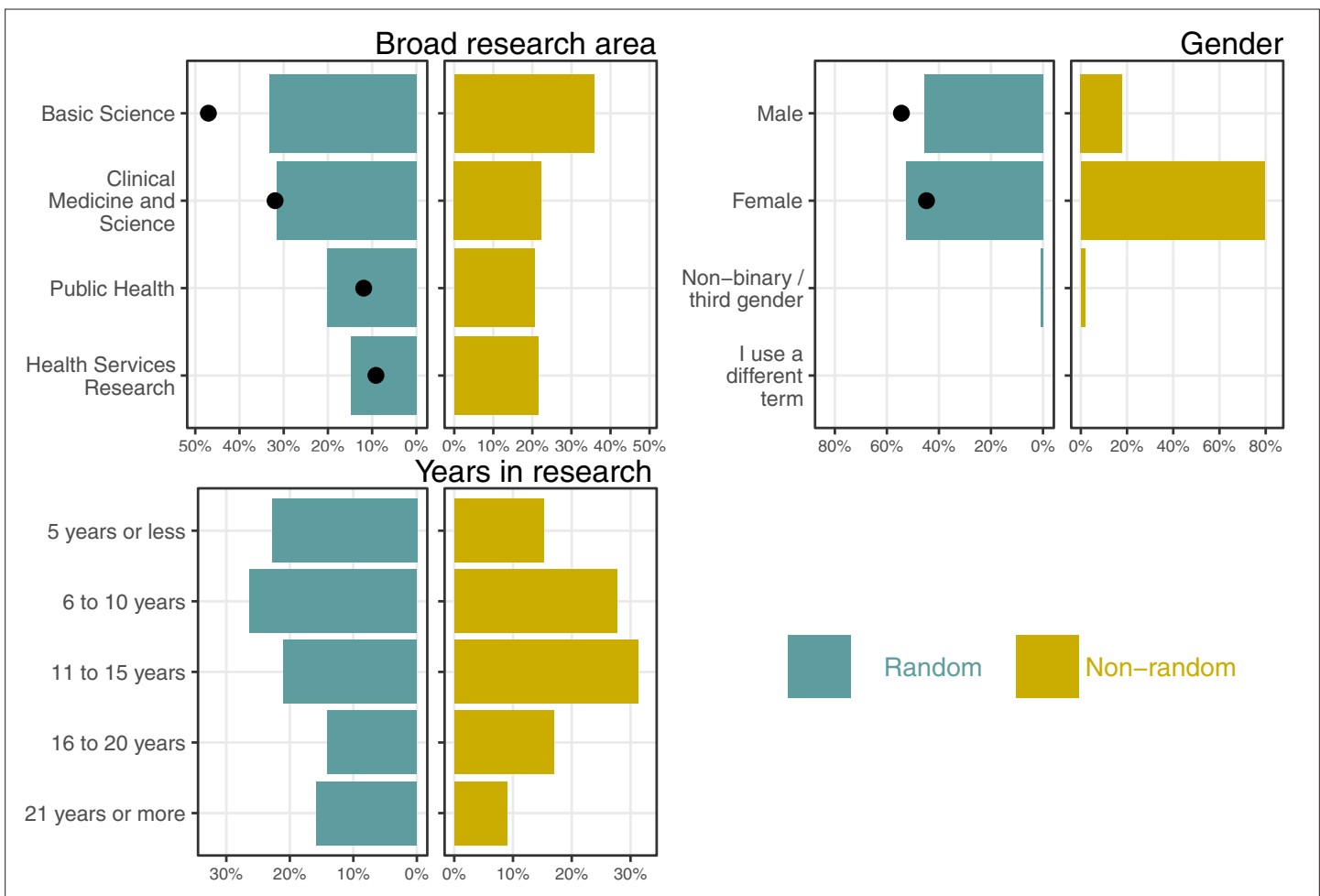

**Figure 1.** Percentages for the three demographic questions split by the random and non-random sample. The black dots are the percentages from the latest available NHMRC data on applicants which are compared with the random sample.

to whom career disruption is relevant. The survey was trapped by at least two spam filters and one respondent initially thought our email approach was a scam. The flow diagram of participants for the random sample is in *Supplementary file 2*, p. 20. The number of respondents to the random (124) and non-random samples (122) was similar.

The survey took a median of 8 min to complete, first to third quartile 6–14 min. Missing data were relatively small and 92% of participants completed 75% or more of the survey. The highest amount of missing data were for the 'slider' questions and this may have been because respondents agreed with the default answer of 6 months, however respondents needed to move the slider for their answer to be recorded. A detailed report on the missing data is in *Supplementary file 2*, pp. 22–25.

## Demographics

A graphical summary of the demographic characteristics are in *Figure 1*. There was a large difference between the random and non-random sample in terms of gender, as 53% of responders in the random sample were women compared with 79% in the non-random sample. There were also more women in the random sample compared with recent NHMRC data on applicants, where 45% were women. These differences are likely because women are generally more motivated than men to discuss career disruption.

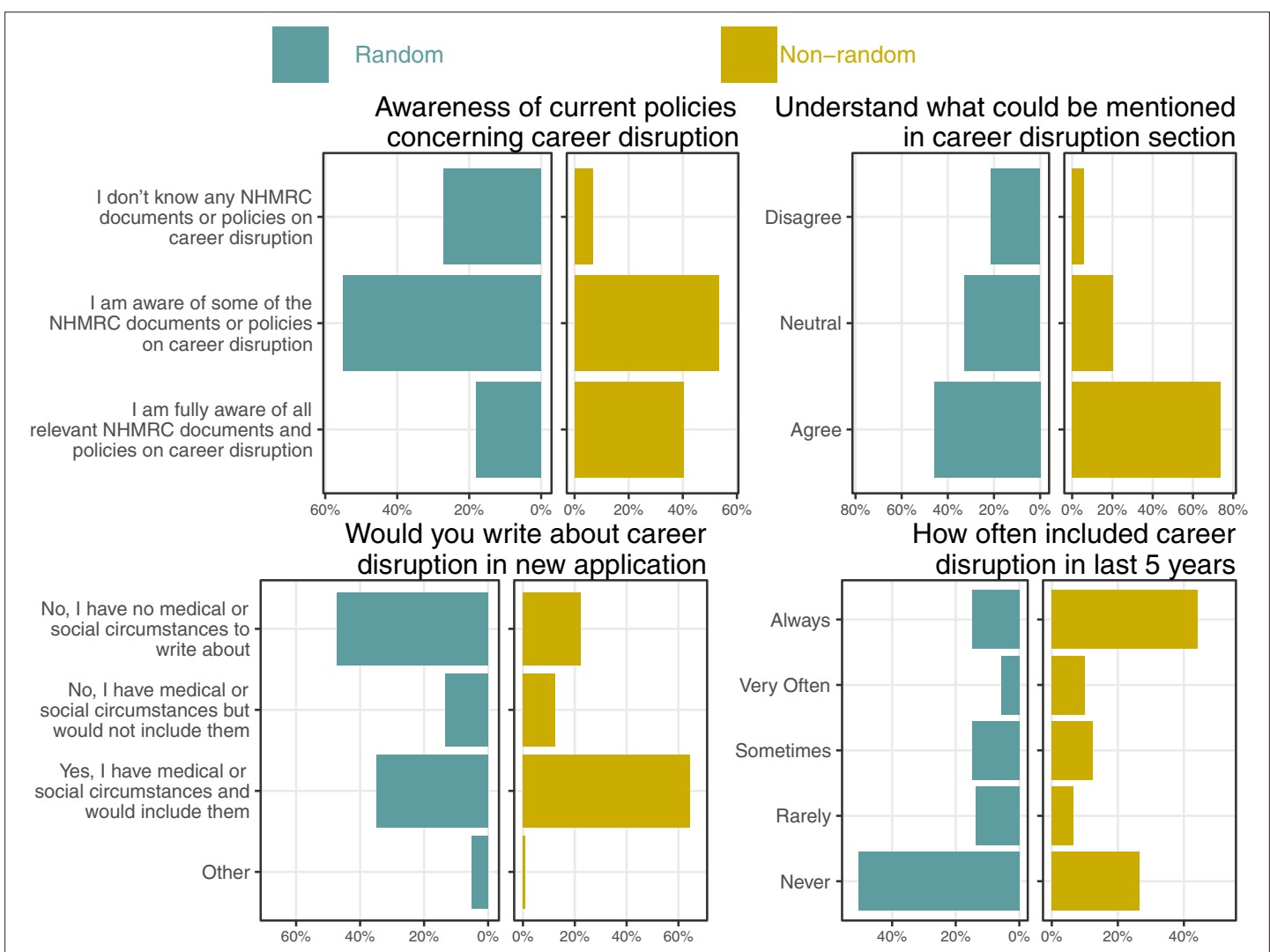

**Figure 2.** Percentages for the questions on awareness and understanding of current career disruption policies and two questions on the respondents' use of career disruption sections split by the random and non-random sample.

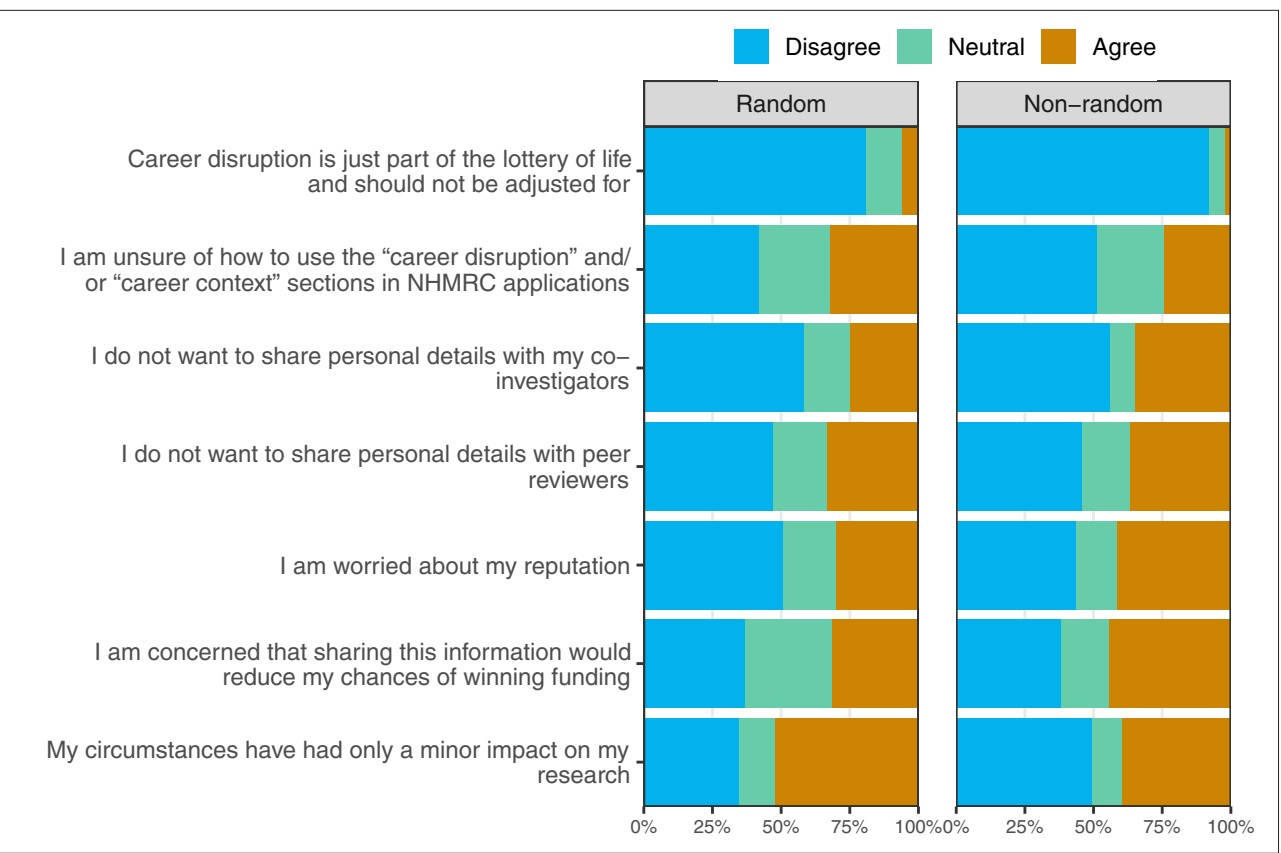

**Figure 3.** Percentages for the reasons given for not including career disruptions sections split by the random and non-random sample. The reasons are ordered by the overall numbers agreeing across the two samples.

We had responses from researchers across all four broad research areas, with more respondents in basic science although this area was under-represented compared with recent NHMRC applicant data (33% in the random sample compared with 47% in NHMRC applicant data). The spread across experience shows that we captured responses from early career researchers through to highly experienced researchers.

## Experience and knowledge of career disruption

The respondents' awareness and understanding of career disruption policies are summarized in *Figure 2*. The percentage fully aware of policies was relatively low at 18% in the random sample (95% CI 10–28%). The percentage of awareness in the non-random sample was 40%, more than double the random sample. This is likely because this group more often had career disruption and hence needed to be familiar with the policies, which is clear in both the question about what respondents have written in recent applications with a much higher percentage of 'Always' including career disruption (44 vs 15%), and a much higher percentage saying they would include career disruption in a current application (64 vs 35%). These percentages may have been increased by the recent COVID-19 lockdowns across Australia, which meant many researchers had to work from home and juggle work and family life.

In the random sample, 13% (95% CI 4–23%) said they have medical or social circumstances but would not include it in an application. The two most common reasons for not including career disruption were that it had only had a minor impact (52% random 40% non-random) and concern that it would reduce their chances of winning funding (32% random 45% non-random; *Figure 3*). Many comments from respondents spoke about the potential harm of including career disruption:

'Worried that people would review my medical conditions as meaning I couldn't do the project.' (Female, 6–10 years).

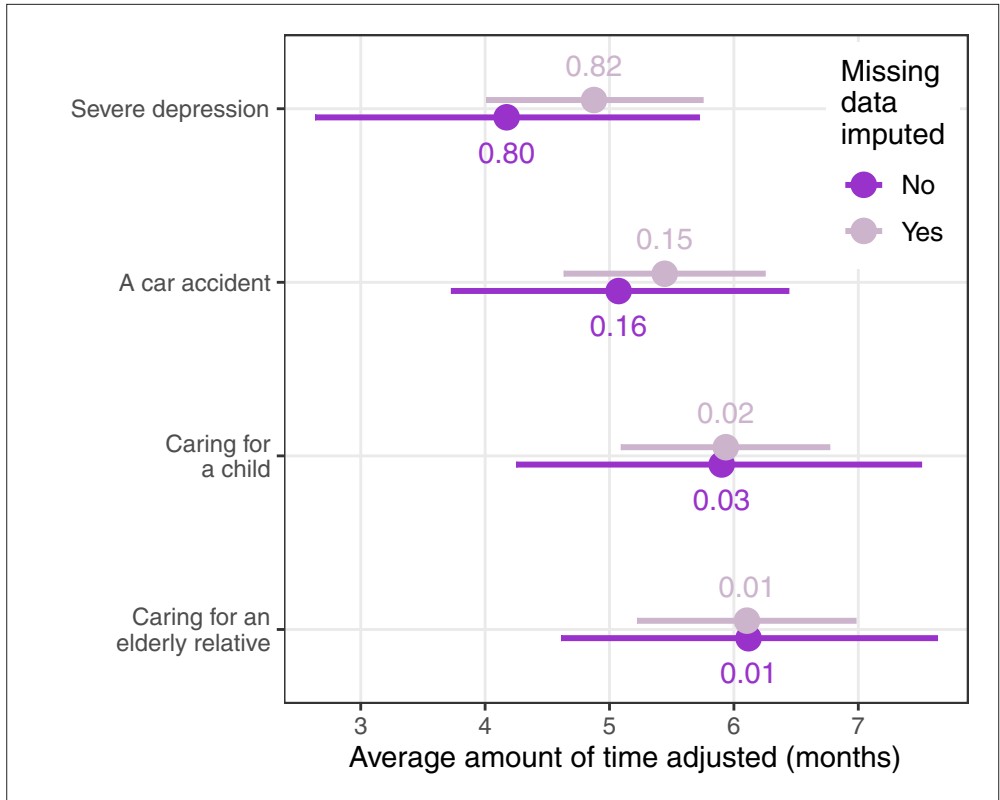

**Figure 4.** Estimated mean time away from research that applicants would write for four hypothetical career disruptions of 6 months duration. The dots are the means and horizontal lines are 95% credible intervals. The numbers above and below the mean are the probability that the scenario had the lowest mean. The results are shown without imputing missing data and imputing 6 months for missing slider data.

'It is such a fine line between making people see just how hard your circumstances have made your academic career (and how hard won your success is), and giving them information you are worried will make the reviewer question your ability to do your research.' (Female, 5 years or less).

A relatively large percentage agreed that they did not want to share personal details with their colleagues (25% random, 35% non-random) or peer reviewers (33% random, 37% non-random). Two quotes that address this are:

'It would be difficult to disclose mental illness or family violence unless this could be done discreetly.' (I use a different term, 21 years or more).

'Its a noble aspiration but would never include any personal details in my application.' (Male, 6–10 years).

Few participants (6% random, 2% non-random) agreed that career disruption is just part of the lottery of life and should not be adjusted for.

### Hypothetical scenario

The results for the four hypothetical scenarios plotted in *Figure 4* show that severe depression had the lowest mean adjustment (4.2 months) and the highest probability of having the lowest mean of the four scenarios. Caring for a child or elderly relative had a mean adjustment close to 6 months and were unlikely to have the lowest mean (probability under 0.03).

After adjusting for the missing slider data, the mean for severe depression increased to 4.9 months, but it still had the highest probability of being the lowest mean. The credible intervals for the results after imputing missing data are narrower as the means are closer to a more consistent 6 months.

Interesting comments in this section include one respondent who was not sure that caring for an elderly relative was an eligible reason for citing career disruption (it is eligible in the current NHMRC

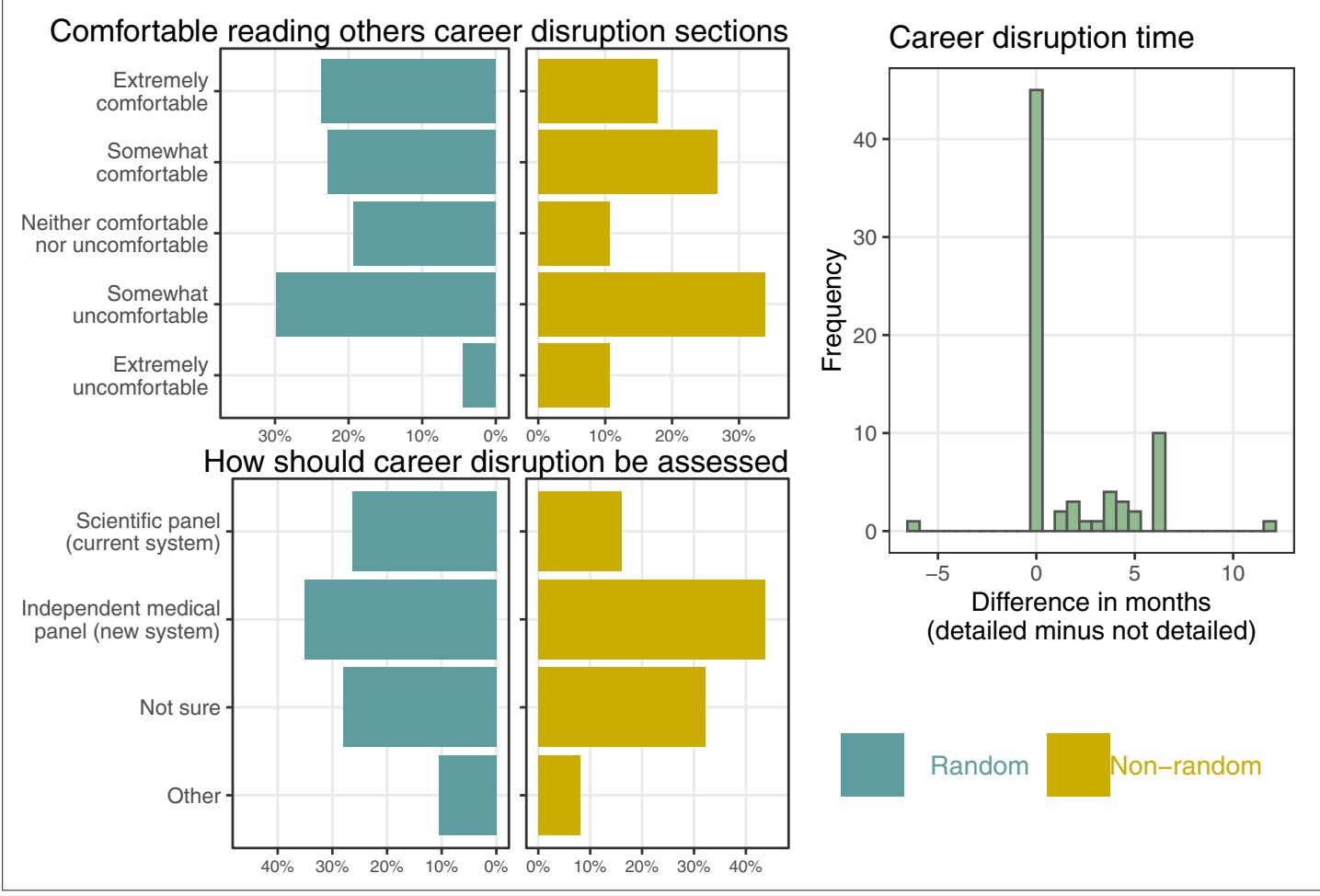

**Figure 5.** Percentages for the three questions concerning the respondents' perspective as a peer reviewer. The histogram shows the difference in time that respondents gave to an applicant who detailed their medical issue and one who did not.

policy), and another who felt that a car accident would 'increase the time I had to write' (Male, 21 years or more).

## Peer reviewer perspectives

Two questions asked respondents to consider their perspective as a peer reviewer, and one asked how they thought career disruption should be assessed in the peer review process. The results are summarized in *Figure 5*.

The respondents were split about whether they were comfortable reading about their colleagues' career disruption sections. A good number were 'extremely comfortable': 24% (random) and 18% (non-random), but 30% (random) and 34% (non-random) said they were 'somewhat uncomfortable'. Using the random sample gives a percentage of 34% being somewhat or extremely uncomfortable with a 95% CI from 25 to 45%. Two quotes on this issue are:

'The applicants have chosen to list disruption. I assume they are comfortable with 5 random strangers reading the information. I, personally, would be happy if not description of the disruption was given but I know many colleagues who would not be and think the applicant is "gaming the system".' (sic, Male, 16–20 years).

'I find all of this really needlessly intrusive. People are entitled to privacy.' (Female, 6–10 years).

Many respondents would give the same length of adjustment for career disruption to those who did and did not provide details about their medical problem. However, where there was a difference, it was almost always to give more time to the applicant who provided details. On average, respondents gave a longer disruption time to those who detailed their medical problem compared with

those who gave no details. The average difference was 1.6 months, 95% CI 1.0–2.3 months. Adjusting for missing data reduced the average difference to 1.2 months (95% CI 0.9–1.5 months), but still showed a clear difference in the mean.

This is a concern given that most respondents – when answering as applicants – said that when including career disruption they would share only the basic information needed to convey the issue (80% random 83% non-random). Many respondents commented on whether or not to share personal information, including:

'One cannot expect an applicant to provide very personal information, but a reviewer will find it hard to distinguish between an applicant that "plays the system" by claiming a disruption without providing detail and one that has a reason but chooses not to divulge.' (Male, 16–20 years).

'Documentation is required to apply for a career disruption and i don't think you could apply for it without providing some information.' (Female, 6–10 years).

There was consensus that changes were needed to ensure career disruption is more consistently accounted for in assessing applications. However, there were differing views on how this could be best achieved. Many respondents from both samples supported the idea of a separate medical/social panel for assessing career disruption. In the random sample 35% preferred the medical panel (95% CI 26–46%), with slightly more support in the non-random sample of 44%. A relatively large percentage of respondents said they were not sure (28% random sample and 32% non-random sample) indicating that more details are needed about this potential change to the peer review process before they could decide. The suggested benefits of an independent medical/social panel were improved confidentiality for both applicants and peer reviewers, along with more consistent and accurate assessment of the time lost to career disruption. Other respondents were concerned that introducing another bureaucratic process could further undermine applicants' professionalism and trust, and could add to the administration and time required to assess applications. Divergent quotes on this issue are:

'The medical review panel would be beneficial in determining the equivalent time lost due to chronic illness or smaller disruptions over extended time frames.' (Male, 16–20 years).

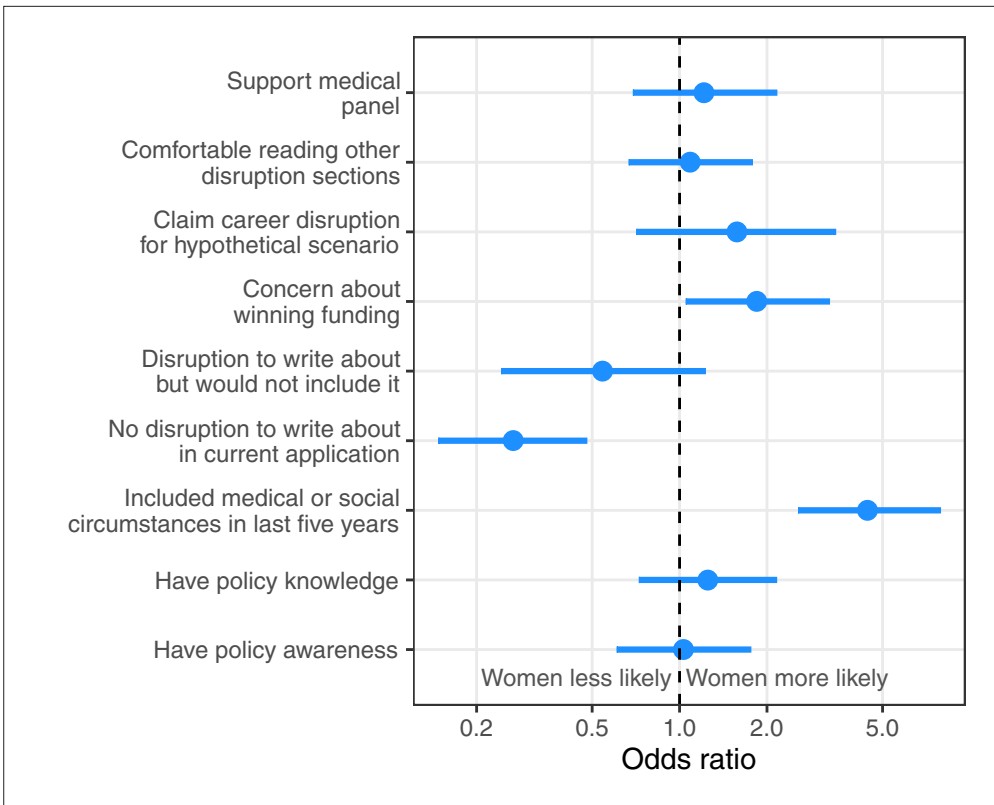

**Figure 6.** Odds ratios to examine if women were more or less likely to agree to selected questions. The odds ratio is on a log-10 scale. An odds ratio of 1 indicates no difference for women relative to men. The horizontal lines are 95% confidence intervals.

'Having a separate external panel scrutinize the claim may create stigma around career disruptions as it implies that claims might not be accurate.' (Non-binary, 11–15 years).

Some respondents made other suggestions for how the current system could be improved, including: more training and clearer guidelines for peer reviewers; a randomized audit of funding decisions to better ensure reviewers consistently account for career disruption; and standardized times for various issues, although some felt that issues are too complex to be standardized. Others suggested assessments could be completed at an organizational level through applicants' human resource departments or by external review once only and then be used for all subsequent applications. The following quotes illustrate the diversity of ideas:

'There needs to be more information and examples for assessors, and there needs to be more allowance made for ongoing or follow on effects from disruption which often last for years afterward.' (Female, 16–20 years).

'Perhaps a statement from the applicant's own medical team should be supplied which could also reduce the cost and time for the nhmrc review process?' (Female, 11–15 years).

'This could be done by the employer at the institution level not by another panel driven by the NHMRC.' (Female, 11–15 years).

## Differences by gender

We examined differences by gender by examining whether women were more likely to answer positively to selected questions. The odds ratios are plotted in *Figure 6*.

Women were much more likely to have had career disruption to write about in past applications, and much more likely to include career disruption in a current application. Interestingly, women were more likely to be concerned about their chances of winning funding if they wrote anything about career disruption (odds ratio 1.8, 95% CI 1.3–3.3). Perhaps surprisingly women were less likely to have disruption but not include it, possibly because the disruption is often caring for children which many are happy to include; although we note that the CI for this association included 1 meaning that there may be no difference for women relative to men. There were no clear differences for the other questions, suggesting attitudes to these issues do not differ greatly by gender. Two quotes on women's perspective are:

'Most career disruption statements are maternity leave related and not a confidentiality concern.' (Female, 11–15 years).

'I am female and don't want to appear "weak". Obvious career breaks such as childbirth, childcare or caring for a partner with cancer are fairly straight forward but other circumstances such as a breakdown could affect my reputation or bias the reviewers to think I would not be able to complete the project.' (Female, 21 years or more).

## Discussion

We surveyed Australian health and medical researchers and used two samples with the aim of getting a representative random sample and using a non-random sample to hear from those keen to speak on this issue. The difference between the non-random and random samples was revealing, with far more women in the non-random sample as women are more often impacted by career disruption. Women continue to be under-represented in NHMRC funding at the senior levels and the fact they more often experience career disruption is a likely reason for this funding gap.

A relatively large percentage of respondents said they have medical or social circumstances but would not include them (13%, 95% CI 4–23%). A common reason for not disclosing was the sense they could be disadvantaged for doing so. Some researchers worry about appearing 'weak' in the words of one female respondent, or risk making a reviewer doubt their long-term capability. This was particularly the case for respondents who had chronic illnesses, mental health issues, or disabilities. If applicants who do not include their career disruptions are taking the best approach, then there is unfairness in the system as those who include career disruption are being disadvantaged. On the other hand, if applicants who do not include disruptions are mistaken, then they are disadvantaging themselves. The reality is likely a mix, as career disruptions will be assessed depending on the peer reviewer, and hence it will sometimes be of benefit to include career

disruption and sometimes not. This variability is likely a contributor to the uncertainty in peer review, where funding decisions depend on the 'luck of the draw' of reviewers (*Graves et al., 2011*).

We found disadvantages for those who are willing to include career disruption both in terms of what their disruption concerned and how much detail they included. In a hypothetical scenario, applicants with the same length of career disruption were (on average) not treated equally, with those experiencing severe depression claiming less time adjustment than those caring for children or elderly relatives. This difference suggests that some peer reviewers will apply their own views of medical conditions. As one respondent commented, the difference could be due to the stigma of mental illness:

'Mental health is incredibly stigmatised and people inadvertently associate major depressive disorder with emotional weakness or lack of resilience. The academic & research sector is particularly stigmatising towards this. I would worry such information would be used against me and to prove that I as a researcher wouldn't have the ability to complete the research successfully. To be honest, if I had to take 6 months off research I probably would never return – any time spent away would be completely uncompetitive.' (Female, 6–10 years).

This quote also highlights how the competitiveness of the funding system influences researchers' actions.

The average difference in adjustment was relatively small at just over 1 month less for applicants with depression compared with those with caring responsibilities. However, few applicants would experience this average as they would either be treated equally or get a larger penalty depending on the attitude of their reviewers. Hence, there is the potential for these differences to influence funding decisions depending on the 'luck of the draw' of reviewers, especially as decisions are often on a knife-edge (*Osmond, 1983*).

In a related issue, applicants who shared details about their medical condition were given a greater time adjustment by peer reviewers (on average) than those who gave no details. This is a concern given that current NHMRC policy states that applicants do not have to provide details and most survey respondents said they would not provide details. Some peer reviewers are unwilling to take the applicants' word for career disruption within the current policy, so including statements like, 'I have missed six months of research in the last five years' with no details on the cause of the disruption could be disadvantaging applicants. Other respondents were concerned that unscrupulous applicants could 'game the system' and over-claim the extent of any career disruption. This reinforces the need for a more transparent and equitable approach to the assessment of career disruption.

Having career disruption also creates an additional burden with respect to paperwork. Applicants with no career disruption do not need to be familiar with the policies or complete the sections of the form. As one respondent said:

'Finding information on the NHMRC is difficult, there is so much information to wade through.' (Female, 21 years or more).

Completing funding applications is time consuming (*Herbert et al., 2013*), often stressful and can conflict with family responsibilities (*Herbert et al., 2014*). It could be particularly stressful for those with career disruption given the conflicting views around what to write and the thought that writing anything might harm their chance of winning funding. For researchers who have experienced career disruption due to a traumatic event or mental illness, even the act of writing about their experience could be exceedingly difficult. One respondent said:

'I would share the information but would really appreciate not having to. It leaves you feeling vulnerable and having to relive the events.' (Female, 11–15 years).

Funders need to consider how to make it easier for those who have experienced disruption to easily document their case (*Jebsen et al., 2019*).

We asked about a potential change to addressing career disruption, using an independent medical/social panel of experts. Most respondents in both samples were supportive of this change, however a relatively large percentage were uncertain (*Figure 5*). A specialist panel could standardize the evaluation of disruption, making it less dependent on the 'luck of the draw' of peer reviewers, and this variability is clear in the time respondents gave for career disruption (*Figure 4*). A respondent commented on this variability:

'Peer-review comments surrounding track record have been so varied over the past 5 years that I do not believe the review system can be trusted to appropriately judge medical conditions.' (Male, 16–20 years).

Using qualified health professionals to judge career disruption would likely be more appropriate compared with the current system where non-medically qualified reviewers, such as statisticians, are tasked with judging potentially complex medical conditions. An independent panel might encourage those applicants currently withholding their career disruption to include it, as their case would be more confidential. Increasing confidentiality would also avoid the need for colleagues to read about each other's personal lives, and we found that many researchers were uncomfortable with sharing this information (*Figure 3*) and reading personal medical details as a peer reviewer (*Figure 5*). A qualified panel that could assess details of the disruption may allay the concerns of some reviewers around applicants gaming the system. Another change that would increase confidentiality would be a system that asked about the impacts of the career disruption rather than its causes.

An alternative system is that peer reviewers provide their scores without considering career disruption, but then a post-assessment adjustment to scores is made dependent on the disruption score from by the medical/social panel. For applicants with no disruption, no change is made, while those with disruption would have their scores increased. The scale of this adjustment would need to be carefully considered. This system removes the task of assessing disruption from peer reviewers and maintains confidentiality. It would not take extra time as the two separate assessments could be made in tandem.

Many respondents suggested that the introduction of another process would not fix the problem of people not disclosing details about their issues, and would cost money and time. There were suggestions to develop more detailed and standardized information about career disruption time and providing that to all reviewers to apply in their judgements. However, this would put additional burden on peer reviewers and it is not clear if reviewers currently read the instructions in detail. A previous study in Australia of the impact of caring for children on research output found there was no 'average impact' of caring for children and that impacts can be variable (*Sewell and Barnett, 2019*). Hence, creating guideline adjustment times for each career disruption reason may be easy to administer but inaccurate.

## Related studies

Our finding that applicants do not share their career disruption as they are concerned about harming their chances of success agrees with a study which found that the pressure to appear 'excellent' makes it harder to disclose health issues (*Brown and Leigh, 2018*). Our finding also agrees with an analysis of academics applying for tenure in the USA (*Pribbenow et al., 2010*), which found stigma about using the 'Stopping the tenure clock' process to adjust for career disruption, with over 20% saying they thought stopping the clock would be viewed negatively.

Recent studies in Australia have documented the disadvantages of being a female scientist. A survey of early career researchers found that young mothers on parental leave continued to write publications while on leave out of fear of falling behind (*Christian et al., 2021*). A survey of scientists' income in all fields found salaries for women were 17% lower than men (*Professionals Australia, 2021*).

The pandemic may have increased the disadvantage for women, with women publishing less (*Squazzoni et al., 2021*; *Vincent-Lamarre et al., 2021*). This is likely because women had to do more home schooling and caring for family members during the pandemic (*Derrick et al., 2021*). Given the importance of track records for winning funding, this makes it even more important that career disruption sections be properly assessed. A recent idea that could reduce the disadvantage for women is for men and women to be awarded in separate funding pools (*ManelWatchAu, 2021*), hence the top-scoring women will always be funded and the gender balance could be easily controlled by the funder. However, this approach could favour women without career disruption and additional policies would be needed concerning career disruption.

A longitudinal study in the USA found that new parents were less likely to stay in full-time STEM employment after their first child compared with those with no caring responsibilities, with a 23% loss for new fathers and 43% loss for new mothers (*Cech and Blair-Loy, 2019*). A study that used focus groups of academics at all career stages found strong support for levelling the playing field for

applicants with a non-traditional career path (including women taking maternity leave) because their track record may have gaps (*McAllister et al., 2015*). Suggested changes included blinding applications to reduce bias and reducing the importance of track record. The 'leaky pipeline' metaphor is often used to describe the lack of women and minorities at senior academic levels, but a 'hostile obstacle course' has been suggested as a more appropriate metaphor (*Berhe et al., 2021*).

An observational study compared two funding rounds in Canada that provided rapid funding during the COVID-19 pandemic (*Witteman et al., 2021*). It found a large increase in applications from women after applicants were allowed to included a one-page statement on the impact of COVID-19, although other changes also took place between the two rounds, including a longer application time and shorter CV requirements.

Previous research has discussed how women more often opt out of their career before it even begins because of the uncertainty of future career disruption (*Ysseldyk et al., 2019*). A female respondent in our survey said that they would opt out of the system if they had a 6-month gap in their career as they would no longer be competitive. A perceived lack of empathy of the funding system for accounting for career disruption could be one of the reasons behind the shortfall of senior women in health and medical research in Australia.

A study of discrimination in hiring practices found that adding structure to selection processes reduced bias in terms of racial discrimination (*Wolgast et al., 2017*). Hence, a potentially greater structure in the assessment of career disruption may increase equity in funding decisions, although care needs to be taken that greater structure does not mean more paperwork for those with disruption. A medical/social panel could be seen as adding more formal structure to the assessment of career disruption.

We checked if international funding agencies had similar policies for accounting for career disruption. The UK Medical Research Council allow applicants to explain any breaks in employment or publication record, with special reference to the pandemic (*UKRI, 2022*). For the Canadian Institute of Health Research, researchers can describe how disruption has impacted on their research in their application and common CV (*CIHR, 2022*). The US National Institute of Health allow explanations of how personal circumstances may have delayed an individual's transition to an independent career or reduced their scientific productivity (*NIH, 2011*). How these disruption sections are used in practice by applicants and reviewers would need to be examined using country-specific research.

## Limitations

This is the first exploration of the important issue of career disruption and there is considerable scope for further study of the motivations of researchers about what to include and when, especially as this is a complex, multi-faceted issue.

Our survey did not capture how ethnic background impacts researchers' perspectives and experiences of career disruption. This is a task for further research.

The sampling frame from *PubMed* included some researchers not working in health and medical research, and hence not part of our target population, for example researchers working in education. We could have narrowed the sampling frame by excluding non-medical journals, although the large number of journals would make this an extensive exercise. There were also four overseas researchers wrongly included in the sample. Our target population was current researchers but this excluded researchers who had left the field, including those who may have left for career disruption reasons.

We did not collect data on our respondents' funding success nor their experience with funding peer review systems, but attitudes to career disruption may depend on success and experience.

Our survey relied on self-reported intentions and reactions to hypothetical scenarios. Behaviour in the real system may differ from the intentions expressed in this survey. An ideal study to estimate if those with career disruption are being disadvantaged is a parallel randomized experiment where reviewers are randomly allocated to one of two versions of the same application, where one includes career disruption and the other not. However, funding agencies rarely use experimentation to test alternative systems (*Guthrie et al., 2017*).

A number of respondents commented that they do not trust the peer review system to adjust for career disruption, for example:

'By and large the competitive nature of these funding schemes mean that assessors cannot make the value based adjustment of scores when faced with very compelling competing grants.' (Male, 16–20 years).

Some indicated this was because of the difficulty of the task, and others because they do not believe reviewers have any sympathy. In hindsight, a question on this issue would have been useful for informing future policy.

## Conclusion

Health and medical researchers with career disruption can be disadvantaged by the current funding system, either because they are unwilling to share their disruption, or because their disruption is not properly accounted for, and also because of the additional time and stress needed to consider and document their disruption. The pandemic has meant more scientists have suffered career disruption and this could increase the empathy for all those with career disruption (*Pourret, 2020*). Given the growing importance of career disruption, and its likely importance in the gender funding gap, funders need to consider how researchers can easily and discretely share their disruption and have it correctly assessed.

## Acknowledgements

Thanks to the National Library of Medicine for making their data available for research. Thanks to Professor Kypros Kypri for his ideas on adjusting for career disruption.

## Additional information

### Competing interests

Adrian Barnett: receives funding from the NHMRC and is a member of the NHMRC Research Committee. Susanna Cramb: receives funding from the NHMRC. The other authors declare that no competing interests exist.

### Funding

| Funder | Grant reference number | Author |
|---|---|---|
| National Health and Medical Research Council | APP1117784 | Adrian Barnett |
| National Health and Medical Research Council | APP2008313 | Susanna Cramb |

The funders had no role in study design, data collection, and interpretation, or the decision to submit the work for publication.

### Author contributions

Adrian Barnett, Conceptualization, Formal analysis, Methodology, Project administration, Software, Writing – original draft; Katie Page, Conceptualization, Investigation, Writing - review and editing; Carly Dyer, Conceptualization, Formal analysis, Methodology, Writing - review and editing; Susanna Cramb, Conceptualization, Investigation, Validation, Writing - review and editing

### Author ORCIDs

Adrian Barnett  http://orcid.org/0000-0001-6339-0374

### Ethics

Ethics approval was obtained from the Queensland University of Technology human research ethics committee. All participants provided informed consent before completing the survey.

### Decision letter and Author response

Decision letter https://doi.org/10.7554/eLife.76123.sa1
Author response https://doi.org/10.7554/eLife.76123.sa2

## Additional files

### Supplementary files
- Transparent reporting form
- Supplementary file 1. Participant information sheet and survey questions.
- Supplementary file 2. Summary results and plots, and examination of missing data.

### Data availability
All data and code are openly available here https://github.com/agbarnett/career_disruption, (copy archived at swh:1:rev:555bffb51ede3af1511a4707ce35aec87785caa2).

The following dataset was generated:

| Author(s) | Year | Dataset title | Dataset URL | Database and Identifier |
|-----------|------|---------------|-------------|------------------------|
| Barnett AG | 2021 | Survey of career disruption | https://github.com/agbarnett/career_disruption/tree/main/data | Github, agbarnett |

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

## Appendix 1

### Publications types that were included and excluded
The full list of types is available on the *PubMed* site here: https://pubmed.ncbi.nlm.nih.gov/help/#publication-types.

### List of included publication types

| Adaptive clinical trial | Evaluation study |
|---|---|
| Case Reports | Journal Article |
| Classical Article | Meta-Analysis |
| Clinical Study | Multicenter Study |
| Clinical Trial | Observational Study |
| Clinical Trial Protocol | Observational Study, Veterinary |
| Clinical Trial, Phase I | Pragmatic Clinical Trial |
| Clinical Trial, Phase II | Randomized Controlled Trial |
| Clinical Trial, Phase III | Review |
| Clinical Trial, Phase IV | Study Characteristics |
| Clinical Trial, Veterinary | Systematic Review |
| Comparative Study | Twin Study |
| Controlled Clinical Trial | Validation Study |
| Equivalence Trial | |

### List of excluded publication types

| Address | Legislation |
|---|---|
| Autobiography | Letter |
| Bibliography | News |
| Biography | Newspaper Article |
| Clinical Conference | Overall |
| Collected Works | Patient Education Handout |
| Congress | Periodical Index |
| Consensus Development Conference | Personal Narrative |
| Consensus Development Conference, NIH | Portrait |
| Dataset | Practice Guideline |
| Dictionary | Preprint |
| Directory | Publication Components |
| Duplicate Publication | Publication Formats |
| Editorial | Published Erratum |
| Electronic Supplementary Materials | Research Support, American Recovery and Reinvestment Act |
| English Abstract | Research Support, N.I.H., Extramural |
| Expression of Concern | Research Support, N.I.H., Intramural |

*Continued on next page*

*Continued*

| Address | Legislation |
|---|---|
| Festschrift | Research Support, Non-U.S. Govt Research Support, U.S. Govt, Non-P.H.S. |
| Government Publication | Research Support, U.S. Govt, P.H.S. |
| Guideline | Retracted Publication |
| Historical Article | Retraction of Publication |
| Interactive Tutorial | Scientific Integrity Review |
| Interview | Support of Research |
| Introductory Journal Article | Technical Report |
| Lecture | Video-Audio Media |
| Legal Case | Webcast |

