## [Editor Report]

This study examined the reporting of career disruption for government funding applications by the Australian research community. Through a survey, the authors found that despite a substantial number of respondents having medical or social circumstances that would be considered under career disruption, most researchers expressed concerns about sharing career disruption information with the view that it would harm their chances of being funded. The outcomes highlight the perceived stigma for reporting career disruption and impacting women to a greater degree, as well as the inadequacy of the system to foster transparency probably due to the competitive research culture.

---

## [Decision Letter]

**Decision letter after peer review:**

Thank you for submitting your article "Meta-research: Justifying career disruption in funding applications, a survey of Australian researchers" for consideration by *eLife*. Your article has been reviewed by 3 peer reviewers, including Ilse Daehn as the Reviewing Editor and Reviewer #2, and the evaluation has been overseen by Mone Zaidi as the Senior Editor. The following individuals involved in review of your submission have agreed to reveal their identity: Mathew G Lewsey (Reviewer #1); Catherine M Abbott (Reviewer #3).

Essential revisions:

1) Include information about the funding landscape in Australia. The average grant size, percentage awarded vs applications, number of applications, gender and age of researchers when awarded their first grant, other support available for young investigators such as institutional support or foundation grants.

2) Clarification on whether the authors are describing Career Disruptions in the strict sense (as described by the NHMRC policy) or the Relative to Opportunity sense, i.e., do not qualify under the strict definition and are considered as part of an applicant's achievements Relative to Opportunity.

*Reviewer #1 (Recommendations for the authors):*

In this work the authors aimed to understand how Australian researchers use the opportunity to disclose career interruptions in grant applications, why they use it in the way they do, and the potential effects these decisions might have on allowances that reviewers make for career interruptions. They also propose potential alternative approaches that might address some of the current system's shortcomings.

I find the results well-written and accessible to a non-specialist. The authors start with a description of the demographics of the respondents and of the difference between the random and non-random samples, so that readers can understand the context. Data are clearly presented, and the question addressed in each section is well articulated. The methods are similarly well-written, and it is pleasing to see the authors' code is already available on GitHub.

The results give clear and detailed insight into how Australian researchers interact with their opportunity to describe career interruptions in funding applications. They first detail the knowledge and experience of respondent's regarding career disruption policies of the funding provider (the NHMRC), which approaches the subject primarily from the perspective of an applicant. They then examine hypothetical scenarios of the time applicants might claim for a specified set of career interruptions. Next, they address attitudes towards career interruptions from the perspective of peer reviewers, which illustrated some general discomfort of reviewers in assessing career interruptions and support for independent expert panels dedicated to this task. This section also demonstrated that generally speaking, applicants are likely to receive a larger time adjustment for career interruptions if they provide details. However, several sections of the study make it clear that applicants are frequently uncomfortable providing such details and explain some of the reasons why they feel so. Lastly, the authors examine whether there are differences by gender in the likelihood that respondents have experienced career interruptions, whether they would claim these in applications, their knowledge of and feelings about doing so, and their support for alternative mechanisms of assessment.

I find that this work has good potential for broad impact in the scientific community. From the viewpoint of individual applicants, it allows them to understand how claiming career interruptions might affect their chances of funding success and make informed decisions about the way that these claims should be presented in applications (for example, how much detail should be included). The authors set their findings in the context of work in other countries, and I find it likely the results are generalisable to other countries' research funding schemes that have similar opportunities to claim career interruptions. More broadly, the manuscript gives detailed information that could be used to improve current systems for describing career interruptions and accounting for them in research funding application schemes, so as to improve equity and the experiences of applicants and reviewers.

Under NHMRC policy, there are Career Disruptions, which have a strict definition. And then there are career disruptions that do not qualify under the strict definition so are considered as part of an applicant's achievements Relative to Opportunity. These then have different implications regarding whether time is added to an applicant's eligibility for certain schemes, versus simply assessing an applicant's achievements relative to others when ranking proposals. Researchers who have never applied for NHMRC (ie. the vast majority of *eLife* readers) will be unaware of this, and I must admit that I found it difficult to completely understand whether the authors were talking about Career Disruptions in the strict sense or the Relative to Opportunity sense. I think it is important this is clarified.

This is a well-written manuscript.

*Reviewer #2 (Recommendations for the authors):*

Academia today must, and is in some way, undergoing the recognition that disruption is now "normal", funding organizations and research institutions should be rethinking the paradigms that influenced their funding decisions and promotion decisions. This study provides insights into the Australian funding system, which allows reporting career disruptions in grant applications. The findings also expose the culture, which collectively supports not to report career breaks given the perceived stigma of appearing "weak" to the reviewers. The study also highlighted a potential issue of concern that may lead to unfairness, as those who include career disruption are being exposed and at a disadvantage as well as those impacted by career disruption who do not report, could further feed the ongoing funding gap for women in Australia, albeit individual or structural factors are difficult to disentangle. However, given the recognition that the COVID-19 pandemic has forced organizations and researchers to adapt, to be more resilient, and to a large extent be more determined to get through the uncertainty and chaos, renders these as flawed views and beliefs. Outcomes from the survey make a clear call for a more transparent, less cumbersome and equitable approach to reporting career disruptions.

It is unclear from the data presented whether the researchers surveyed were funded, unfunded and seeking funding. Or whether any of them participate or have participated in funding review panels. If not determined, this should be included in the limitations.

Because these concerns are not unique to Australian researchers, a scholarly comparison with other countries on how career disruptions are handled is needed in the Discussion section.

The discussion could include the 'gender equality paradox' in Sweden, where extensive family policies appear to discriminate and hamper women's advancement in STEM careers that require continuous presence and skill development.

The authors report the shared views among health and medical researchers that reporting career disruption could be disadvantageous by the current funding system in Australia. The results suggest that this is due to the unwillingness to share the disruption, or because the disruption is not properly accounted for, or the additional time and stress needed to consider and document the issues.

In the introduction section the authors should introduce the funding landscape in Australia. If available, it would be interesting to know the average grant size, percentage awarded vs applications, number of applications, gender and age of researchers when awarded their first grant, other support for young investigators.

*Reviewer #3 (Recommendations for the authors):*

This study addresses the important and often neglected issue of how to adjust for career-disrupting events when assessing past records in grant applications. The main funding body in Australia for biomedical research, NHMRC, has a system that allows people to declare any personal circumstances that have impacted their research, and instructs peer reviewers to take these into account. A survey was conducted on attitudes of both applicants and reviewers using two populations, one self-selected ("non-random") and one selected from Pubmed authors ("random"). Results are displayed by gender and research area. Answers given to one of four hypothetical scenarios assigned at random are displayed in terms of number of months of allowance that would be made. The weakness of the study is the relatively small sample size (124 non-random and 122 random participants) and the focus on a single funding system, but it is hard to envisage an in-depth study that encompassed multiple funding bodies worldwide. In many ways the strength of this study is the qualitative results, including quotations, as these are so clearly generalisable beyond the Australian funding system. Data are very clearly presented. There is also an interesting and valuable discussion placing results in the context of related studies in other countries, though this section seems more focused on gender than the results (which cover issues like mental illness and calamities) and contains references to quite a number of blog posts rather than to-peer-reviewed literature.

I am an experimental scientist with no expertise in survey design or analysis so have only one specific comment and that would be to make it clear in the main text that there are almost identical numbers in the random and non-random survey groups.

---

## [Author Response]

Essential revisions:1) Include information about the funding landscape in Australia. The average grant size, percentage awarded vs applications, number of applications, gender and age of researchers when awarded their first grant, other support available for young investigators such as institutional support or foundation grants.

We have added this information to a new paragraph in the Introduction. To our knowledge, Information on winners’ ages is not shared by the NHMRC. The new paragraph is:

“The National Health and Medical Research Council (NHMRC) are the largest government funding agency for health and medical research in Australia. In 2020--21 there were 536 grants awarded with a total budget of $497 million AUD (NHMRC, 2021a). Across all schemes the success percentages by the gender of the lead investigator were 13.0% for women and 12.9% for men, but were 12.7% (women) and 14.0% (men) for Investigator grants (akin to fellowships). For the lead investigators for all schemes, there were 1,678 applications submitted by women and 2,154 by men. There were 64 postgraduate scholarships awarded to early career researchers. While smaller external and internal schemes are available to researchers, promotion often requires a large grant from the NHMRC or equivalent (Rice et al., 2020).”

2) Clarification on whether the authors are describing Career Disruptions in the strict sense (as described by the NHMRC policy) or the Relative to Opportunity sense, i.e., do not qualify under the strict definition and are considered as part of an applicant's achievements Relative to Opportunity.

We focused on career disruption and have added the following sentence to the methods to clarify this: “Our questions concerned prolonged career disruption rather than "relative to opportunity" which accounts for time spent away from research due to activities like clinical work and teaching.” We have also added a sentence to the introduction on the NHMRC’s “relative to opportunity” section: “There is also a ‘relative to opportunity’ section where applicants can detail other personal or professional circumstances affecting research productivity”.

Reviewer #1 (Recommendations for the authors):[…]Under NHMRC policy, there are Career Disruptions, which have a strict definition. And then there are career disruptions that do not qualify under the strict definition so are considered as part of an applicant's achievements Relative to Opportunity. These then have different implications regarding whether time is added to an applicant's eligibility for certain schemes, versus simply assessing an applicant's achievements relative to others when ranking proposals. Researchers who have never applied for NHMRC (ie. the vast majority of eLife readers) will be unaware of this, and I must admit that I found it difficult to completely understand whether the authors were talking about Career Disruptions in the strict sense or the Relative to Opportunity sense. I think it is important this is clarified.This is a well-written manuscript.

We focused on career disruption. Our wording and scenarios in the questionnaire were around disruption. We have added a sentence to clarify this to the “Survey questions” sub-section of the Methods (see point 2 response to “essential revisions” above).

Reviewer #2 (Recommendations for the authors):Academia today must, and is in some way, undergoing the recognition that disruption is now "normal", funding organizations and research institutions should be rethinking the paradigms that influenced their funding decisions and promotion decisions. This study provides insights into the Australian funding system, which allows reporting career disruptions in grant applications. The findings also expose the culture, which collectively supports not to report career breaks given the perceived stigma of appearing "weak" to the reviewers. The study also highlighted a potential issue of concern that may lead to unfairness, as those who include career disruption are being exposed and at a disadvantage as well as those impacted by career disruption who do not report, could further feed the ongoing funding gap for women in Australia, albeit individual or structural factors are difficult to disentangle. However, given the recognition that the COVID-19 pandemic has forced organizations and researchers to adapt, to be more resilient, and to a large extent be more determined to get through the uncertainty and chaos, renders these as flawed views and beliefs. Outcomes from the survey make a clear call for a more transparent, less cumbersome and equitable approach to reporting career disruptions.

Thank you for this clear interpretation of our paper.

It is unclear from the data presented whether the researchers surveyed were funded, unfunded and seeking funding. Or whether any of them participate or have participated in funding review panels. If not determined, this should be included in the limitations.

We deliberately kept our questionnaire short to increase the response rate and so did not ask these questions. In the limitations, we have added that this information was not collected: “We did not collect data on our respondents' funding success nor their experience with funding peer review systems, but attitudes to career disruption may depend on success and experience.”

Because these concerns are not unique to Australian researchers, a scholarly comparison with other countries on how career disruptions are handled is needed in the Discussion section.

We have checked the application procedures for major funders in Canada, the UK and USA, and have added this information to the conclusion. How disruption is handled in practice would need to be examined using similar surveys in each of these countries. New paragraph:

“We checked if international funding agencies had similar policies for accounting for career disruption. The UK Medical Research Council allow applicants to explain any breaks in employment or publication record, with special reference to the pandemic (MRC, 2022). For the Canadian Institute of Health Research, researchers can describe how disruption has impacted on their research in their application and common CV (CIHR, 2022). The US National Institute of Health allow explanations of how personal circumstances may have delayed an individual’s transition to an independent career or reduced their scientific productivity (NIH, 2011). How these disruption sections are used in practice by applicants and reviewers would need to be examined using country-specific research.”

The discussion could include the 'gender equality paradox' in Sweden, where extensive family policies appear to discriminate and hamper women's advancement in STEM careers that require continuous presence and skill development.

Thanks for raising this phenomenon, which we were not aware of. Reading the paper by Richardson et al. (DOI: 10.1177/0956797619872762) we note their valid criticism that a key paper showing the Gender-Equality Paradox used a relative rather than absolute measure for women’s participation in STEM, looking at the proportion of female graduates relative to men, rather than the overall number of female graduates. Richardson et al. used an absolute measure and found no association between women’s STEM participation and gender equality. We also agree with Richardson et al’s concerns about the gender-equality paradox studies being based on between country analyses that are vulnerable to the ecological fallacy. In our opinion, a good longitudinal study within a country is needed before we can have any confidence that this paradox is a real phenomenon.

The authors report the shared views among health and medical researchers that reporting career disruption could be disadvantageous by the current funding system in Australia. The results suggest that this is due to the unwillingness to share the disruption, or because the disruption is not properly accounted for, or the additional time and stress needed to consider and document the issues.In the introduction section the authors should introduce the funding landscape in Australia. If available, it would be interesting to know the average grant size, percentage awarded vs applications, number of applications, gender and age of researchers when awarded their first grant, other support for young investigators.

We have added this information to the Introduction (see point 1 response to “essential revisions” above). To our knowledge, the NHMRC does not share information on the age of grant winners.

Reviewer #3 (Recommendations for the authors):This study addresses the important and often neglected issue of how to adjust for career-disrupting events when assessing past records in grant applications. The main funding body in Australia for biomedical research, NHMRC, has a system that allows people to declare any personal circumstances that have impacted their research, and instructs peer reviewers to take these into account. A survey was conducted on attitudes of both applicants and reviewers using two populations, one self-selected ("non-random") and one selected from Pubmed authors ("random"). Results are displayed by gender and research area. Answers given to one of four hypothetical scenarios assigned at random are displayed in terms of number of months of allowance that would be made. The weakness of the study is the relatively small sample size (124 non-random and 122 random participants) and the focus on a single funding system, but it is hard to envisage an in-depth study that encompassed multiple funding bodies worldwide. In many ways the strength of this study is the qualitative results, including quotations, as these are so clearly generalisable beyond the Australian funding system. Data are very clearly presented. There is also an interesting and valuable discussion placing results in the context of related studies in other countries, though this section seems more focused on gender than the results (which cover issues like mental illness and calamities) and contains references to quite a number of blog posts rather than to-peer-reviewed literature.

Thanks for this clear review. Our sample size exceeded our target, and so our margin of error for all questions is less than 10% which gives us reasonable confidence in our estimates.

We have updated the discussion and included three more peer reviewed papers and one government report.

– Berhe AA, Barnes RT, Hastings MG, Mattheis A, Schneider B, Williams BM, Marín-Spiotta E. Scientists from historically excluded groups face a hostile obstacle course. Nature Geoscience. 2021; 15(1):2–4.

– Brown N, Leigh J. Ableism in academia: where are the disabled and ill academics? Disability & Society. 2018; 33(6):985–989.

– Cech EA, Blair-Loy M. The changing career trajectories of new parents in STEM. Proceedings of the National Academy of Sciences. 2019; 116(10):4182–4187.

– McAllister D, Juillerat J, Hunter J. Towards a better understanding of issues affecting grant applications and success rates by female academics. Biotechnology and Biological Sciences Research Council. 2015.

I am an experimental scientist with no expertise in survey design or analysis so have only one specific comment and that would be to make it clear in the main text that there are almost identical numbers in the random and non-random survey groups.

We have added this to the first paragraph of the results: “The number of respondents to the random (124) and non-random samples (122) were similar.”